# Programmed Cell Death-1/Programmed Cell Death-1 Ligand as Prognostic Markers of Coronavirus Disease 2019 Severity

**DOI:** 10.3390/cells11121978

**Published:** 2022-06-20

**Authors:** Paulina Niedźwiedzka-Rystwej, Adam Majchrzak, Bogusz Aksak-Wąs, Karol Serwin, Zenon Czajkowski, Ewelina Grywalska, Izabela Korona-Głowniak, Jacek Roliński, Miłosz Parczewski

**Affiliations:** 1Institute of Biology, University of Szczecin, 71-412 Szczecin, Poland; paulina.niedzwiedzka-rystwej@usz.edu.pl; 2Department of Infectious, Tropical Diseases and Immune Deficiency, Pomeranian Medical University in Szczecin, 70-204 Szczecin, Poland; adammajchrzak@protonmail.com (A.M.); bogusz.aw@gmail.com (B.A.-W.); karol.serwin@pum.edu.pl (K.S.); 3Intensive Care Unit, Public Regional Hospital in Szczecin, 71-455 Szczecin, Poland; czajkowski@spwsz.szczecin.pl; 4Department of Experimental Immunology, Medical University of Lublin, 20-093 Lublin, Poland; ewelina.grywalska@gmail.com; 5Department of Pharmaceutical Microbiology, Medical University of Lublin, 20-093 Lublin, Poland; iza.glowniak@umlub.pl; 6Department of Clinical Immunology and Immunotherapy, Medical University of Lublin, 20-093 Lublin, Poland; jacek.rolinski@gmail.com

**Keywords:** immunophenotype, disease severity, severe acute respiratory syndrome coronavirus 2, lymphocyte, programmed cell death protein 1

## Abstract

Current research proves that immune dysregulation is a common feature of coronavirus disease 2019 (COVID-19), and immune exhaustion is associated with increased disease mortality. Immune checkpoint molecules, including the programmed cell death-1 (PD-1)/PD-1 ligand (PD-L1) axis, may serve as markers of disease severity. Accordingly, in this study, we evaluated the expression of PD-1/PD-L1 in patients with COVID-19. Blood immunophenotypes of hospitalized patients with moderate (n = 17, requiring oxygen support) and severe (n = 35, requiring mechanical ventilation in the intensive care setting) COVID-19 were compared and associated with clinical, laboratory, and survival data. The associations between severity and lymphocyte profiles were analysed at baseline and after 7 and 14 days of in-hospital treatment. Forty patients without COVID-19 infection were used as controls. For PD-1-positive T and B lymphocyte subsets, notable increases were observed between controls and patients with moderate or severe COVID-19 for CD4+PD-1+ T cells, CD8+PD-1+ T and CD19+PD-1+ B cells. Similar trends were observed for PD-L1-positive lymphocytes, namely, CD4+PD-L1+ T cells, CD8+PD-L1+ T cells and CD19+PD-L1+ B cells. Importantly, all markers associated with PD-1 and PD-L1 were stable over time for the analysed time points in the moderate and severe COVID-19 groups. Increased abundances of PD-1+ and PD-L1+ lymphocytes were associated with disease severity and mortality and were stable over time in patients with moderate to severe COVID-19. These immune exhaustion parameters may be attractive biomarkers of COVID-19 severity.

## 1. Introduction

Since its discovery in 2019, coronavirus disease 2019 (COVID-19) has killed over 5.7 million people worldwide, with approximately 5% of infections leading to respiratory failure and necessitating intensive care admission and either high oxygen flow or mechanical ventilation [1]. Age, comorbidities, and male sex remain the classical risk factors for the progression of severe disease and mortality [2] and are strongly correlated with inappropriate and excessive immunological activity, also known as a cytokine storm [3]. Treatment success in severely ill patients requiring mechanical ventilation is limited, with early and late mortality rates exceeding 50% when patients are admitted to an intensive care unit (ICU) [4]. The most common causes of death in the intensive care setting include direct respiratory insufficiency, bacterial superinfections, and cardiac failure [5]. Thus, improved therapeutic strategies are urgently needed, and further studies are necessary to investigate and identify the factors associated with the risk of disease progression and ICU admission and to characterise lymphocyte functions among critically ill patients with COVID-19 to identify survival markers and therapeutic targets. 

Immune checkpoint molecules, including programmed cell death-1 (PD-1) and PD-1 ligands 1 and 2 (PD-L1 and PD-L2, respectively), function to protect against the over-reaction of the host’s immune system to infections. As a negative regulatory receptor, PD-1 is expressed on immune cells; under conditions of homeostasis, PD-1 functions in conjunction with PD-L1 and PD-L2 to inhibit the activity of immune system cells via autotolerance to healthy cells of the body [6]. PD-1 is expressed in T and B lymphocytes, natural killer (NK) cells, NK T cells, monocytes, and dendritic cells (DCs). PD-L1 is the ligand for PD-1, which is expressed on the surface of haematopoietic, endothelial, epithelial, and mesenchymal stem cells, as well as pathogenic neoplastic cells. The upregulation of PD-1 protein on T cells and increased PD-L1 expression are associated with T cell depletion, which occurs in patients with chronic viral infection, such as those caused by human immunodeficiency virus (HIV) [7], hepatitis B virus (HBV), hepatitis C virus (HCV) [8,9], herpes simplex virus (HSV) [10], Epstein–Barr virus (EBV) [11], varicella-zoster virus (VZV) [12], and cytomegalovirus [10]. Overexpression of PD-1/PD-L1 is also associated with immune-exhaustion, which occurs when cells are constantly stimulated by antigens or are inhibited by the strong binding of PD-L1 to its receptor, resulting in a decrease in T lymphocyte proliferation and loss of its effector functions, including apoptosis. However, mechanisms regulating the PD-1/PD-L1 axis during viral infection are complex and vary according to the disease and virus type, leading to different phenotypes [13]. 

Therefore, in this study, we evaluated differences in PD-1 and PD-L1 expression according to COVID-19 disease severity by comparing patients requiring ICU admission with less severely ill hospitalized patients. Furthermore, we analysed the stability of the PD-1/PD-L1 signalling axis in patients over time (three 7-day intervals) and assessed correlations between PD-1/PD-L1 expression and selected biochemical parameters.

## 2. Materials and Methods

### 2.1. Study Groups

In this study, we analysed the samples and dataset of 52 randomly selected in-hospital-treated patients with moderate (requiring oxygen support only, referred to as the non-ICU group) or severe (with mechanical ventilation in the ICU, referred to as the ICU group) COVID-19 pneumonia at a 1:2 ratio. All patients presented with clinical symptoms of cough, dyspnoea, or fever (>38 °C), and oxygen saturation less than or equal to 94% prior to hospital admission. In every case, polymerase chain reaction (PCR) for severe acute respiratory syndrome coronavirus 2 (SARS-CoV-2) was performed using pharyngeal swabs confirming infection with this virus, and pneumonia was confirmed using chest computed tomography (CT). We collected samples from patients admitted between December 2020 and January 2021. We wished to study immunologic parameters not only in the context severe disease but also mortality; therefore, we compared immunophenotypes between the surviving group (n = 38) and patients who died of COVID-19 pneumonia (n = 14). 

### 2.2. Ethical Issues

The study protocol was approved by the Bioethical Committee of Pomeranian Medical University, Szczecin, Poland (approval number: KB-0012/92/2020). All patients or their legal representatives provided informed consent for participation in the study, related immunophenotyping procedures, and data collection. Data were collected anonymously. The study was conducted in accordance with principles of the Declaration of Helsinki. 

### 2.3. Sampling and Data Collection Methodology

In this study, we collected clinical data from medical records, including age, sex, treatment history, duration of in-hospital stay, duration of treatment in the ICU, survival statistics, baseline blood oxygenation levels, chest CT scan results, comorbidities, concomitant medications, and selected laboratory parameters (white blood cell count, haemoglobin levels, platelet count, procalcitonin levels, C-reactive protein levels, interleukin 6 levels, lactate dehydrogenase levels, d-dimer activity, and aspartate and alanine aminotransferase activity). Full blood samples for fluorescence-assisted cell sorting (FACS) were collected on admission to the hospital and ICU (baseline) and 7 and 14 days thereafter. 

### 2.4. Immunophenotyping 

For the above-mentioned groups of patients, a detailed immunophenotype analysis of various parameters, such as the frequencies of NK cells and NK-like cells and the expression of individual antigens related to cellular differentiation on T and B lymphocytes, was performed, with particular emphasis on PD-1 and PD-L1. Immunophenotyping results were compared with those from the control group, which consisted of 40 individuals (67.5% men and 32.5% women) without symptoms of SARS-CoV-2 infection, with negative reverse transcription PCR results and medical history free from allergies or immunity deficiency. For immunophenotyping, peripheral blood samples for frequency analysis were collected in ethylenediaminetetraacetic-acid-containing tubes. The cells were examined based on unstained control, FMO control, and stained cells with monoclonal antibodies conjugated with fluorescent dyes as follows: 3-color labelling with the following surface antibodies: mouse fluorescein isothiocyanate (FITC)-conjugated anti-human CD4 (clone SK3)/mouse phycoerythrin (PE)-conjugated anti-CD8 (clone SK1)/mouse peridinin chlorophyll protein (PerCP)-conjugated anti-CD3 (clone SK7) to determine the proportion of the CD4+CD3+ T lymphocytes and CD8+CD3+ T lymphocytes; 2-color labelling with following surface antibodies: mouse FITC-conjugated anti-human CD3 (clone SK7)/mouse PE-conjugated anti-CD16 (clone B73.1), mouse PE-conjugated anti-CD56 (clone MY31) to determine the proportion of the NK and NKT-like cells and mouse FITC-conjugated anti-human CD3 (clone SK3)/mouse PE-conjugated anti-CD19 (clone 4G7) to determine T and B lymphocytes (Becton Dickinson, East Rutherford, NJ, USA). Percentages of PD-1-positive and PD-L1-positive lymphocytes were determined using 2-colour labelling combinations of the following monoclonal antibodies: mouse FITC-conjugated anti-CD19 (clone HIB19)/mouse PE-conjugated anti-PD-1 (clone MIH4), mouse FITC-conjugated anti-CD4 (clone L200)/mouse PE-conjugated anti-PD-1 (clone MIH4), mouse FITC-conjugated anti-CD8 (clone SK1)/mouse PE-conjugated anti-PD-1 (clone MIH4), mouse FITC-conjugated anti-CD19 (clone HIB19)/mouse PE-conjugated anti-PD-L1 (clone MIH1), mouse FITC-conjugated anti-CD4 (clone L200)/mouse PE-conjugated anti-PD-L1 (clone MIH1), mouse FITC-conjugated anti-CD8 (clone SK1)/mouse PE-conjugated anti-PD-L1 (clone MIH1), (Becton Dickinson, East Rutherford, NJ, USA). Cells were incubated with Fc Receptor (FcR) blocking reagent (Miltenyi Biotec, CA, USA) for 10 min at room temperature to block unspecific FcR-mediated binding of antibodies. Next, the cells were incubated for 20 min at room temperature with 20 μL of each mAb per sample for 30 min at 4 °C in the dark. After staining of samples with antibodies, cells were treated with lysing solution (Becton Dickinson) and incubated for 15 min at 4 °C in the dark. The cells were then washed twice with phosphate-buffered saline. Cell subsets were detected using cell labelling and gating methods, which start by removing doublets (FSC-A vs. FSC-H), followed by a dot-plot, in which lymphocyte populations were defined (FSC vs. SSC). The identification of PD-1-positive and PD-L1-positive cells by flow cytometry in healthy controls, in patients with COVID-19 hospitalized not in the ICU and in the ICU is presented in the Appendix A. Appendix A shows FMO control. Appendix A shows the gate strategy of lymphocytes population. The data were collected on an eight-colour FACSCantoII flow cytometer (BD Biosciences, Franklin Lakes, NJ, USA). The Kaluza Analysis program (Beckman Coulter, CA, USA) was used for data analysis, and the percentage of positive cells was recorded. At least 10,000 events were acquired for each sample.

### 2.5. Statistics

Statistical comparisons were performed using Fisher’s exact and χ^2^ tests for nominal variables, as appropriate. Continuous variables were analysed using the Mann–Whitney U test for nonparametric statistics. Confidence intervals (CIs) and interquartile ranges (IQRs) are indicated where appropriate. Kaplan–Meier cumulative mortality was calculated for the selected immunophenotypic factors [14], and the statistical significance of survival data was analysed using log-rank tests [15]. Additionally, the diagnostic effectiveness of the laboratory test was determined using receiver operating characteristic (ROC) curves for parameters related to the different groups of patients. Areas under the ROC curves (AUCs) were calculated for each parameter and compared. Results with p values less than 0.05 were considered significant. Commercial software (Statistica 13.0 PL; Statasoft, Warsaw, Poland) was used for the statistical calculations. 

## 3. Results

The results of the study are collected in Table 1, Table 2, Table 3, Table 4, Table 5, Table 6, Figure 1 and Figure 2.

### 3.1. Clinical Characteristics of Patients with COVID-19 

The analysed group included 35 patients requiring mechanical ventilation admitted to the ICU and 17 patients with moderate disease requiring oxygen support but not mechanical ventilation. Overall, among both groups, 14 (26.9%) deaths were observed, with notably higher mortality in the ICU group (n = 13, 37.1%) than in the non-ICU group (5.9% [n = 1]; *p* = 0.017). The groups were balanced for age and sex. The overall hospitalization time for patients in the ICU group was longer, with a median of 45 days (IQR: 12–101 days) as compared with 17 days (IQR: 11–29 days) in the non-ICU group (*p* < 0.0001). Notably higher activity of inflammatory parameters was also observed in the ICU group (Table 1). The median time from hospital admission to ICU admission was 2 days (IQR: 2–5 days). 

**Table 1 cells-11-01978-t001:** Clinical and baseline laboratory characteristics of the analysed patient groups.

Study Group	Non-ICU (n = 17)	ICU (n = 35)	*p* Value	Total
Mortality, n (%)				
Survived	16 (94.1)	22 (62.9)	0.017	38 (73.1)
Died	1 (5.9)	13 (37.1)		14 (26.9)
Sex, n (%)				
Male	24 (68.6)	13 (76.5)	n.s.	37 (71.2)
Female	11 (31.4)	4 (23.5)		15 (28.9)
Age, median (IQR)	74 (63–92)	69 (59–73)	n.s.	69 (59.5–74)
Days of hospitalization, median (IQR)	17 (15–29)	45 (31–80)	<0.0001	32 (17.5–61.5)
Anti-COVID-19 treatment during in-hospital stay, n (%)				
Remdesivir	11 (31.4)	2 (15.4)	n.s.	19 (36.5)
Tocilizumab	14 (40.0)	5 (29.4)		13 (25.0)
Convalescent plasma	7 (20.0)	2 (11.8)		9 (17.3)
Selected baseline laboratory parameters, median (IQR)				
WBC [cells/μL]	7.36 (4.93–14.9)	11.49 (7.19–14.43)	0.004	9.82 (6.11–13.43)
NEU [cells/μL]	5.8 (3.3–13)	9.4 (5.9–12.7)	0.004	7.7 (4.25–11.4)
CRP [mg/L]	49.39 (26.42–241.94)	107.09 (41.12–195.92)	0.05	80.78 (29.77–158.995)
IL-6 [pg/mL]	36.5 (22.6–454)	122 (40.3–335)	0.037	83.4 (28.65–206.5)
LDH [U/L]	308 (246–528)	567 (321–690)	0.0002	418.5 (304.5–602)
D-dimer [µg/L]	583 (323–60102)	1547 (856–2790)	0.031	1333 (496.5–2309)
Creatinine levels [mg/dL]	0.94 (0.88–11.85)	0.85 (0.61–1.12)	n.s.	0.9 (0.74–1.17)

IQR, interquartile range. n.s., not significant. All patients received dexamethasone.

### 3.2. Immunophenotype Differences between the Analysed Groups and Controls 

Next, we compared the baseline immunophenotype parameters of COVID-19 ICU and non-ICU groups with healthy controls (Table 2). Among controls, higher percentages of NK cells and CD8+ T lymphocytes were observed compared with either COVID-19 ICU-hospitalized and non-ICU groups. Furthermore, the lymphocyte CD4+/CD8+ ratio was the highest in the COVID-19 ICU group versus in the control group, which was directly related to the lower CD8+ lymphocyte count among severely ill cases. There were no additional differences in the overall percentage distribution of NK, NK-like, CD4+ T, and CD8+ T lymphocytes between the control and COVID-19 groups or between COVID-19 ICU and non-ICU patients. 

**Table 2 cells-11-01978-t002:** Immunophenotypes of COVID-19 cases at baseline (day 0 of in-hospital treatment) compared with healthy controls.

Characteristics		Patients with COVID-19 Hospitalized in the ICU	Patients with COVID-19 not Hospitalized in the ICU	Healthy Controls	*p* Values			Entire COVID-19 Group
		Median (IQR)	Median (IQR)	Median (IQR)	ICU Versus Controls	Non-ICU Versus Controls	ICU Versus Non-ICU	Median (IQR)
Frequencies of individual cells (%)	CD3^−^CD16^+^CD56^+^ NK cells	12.85 (6.63–15.96)	12.22 (10.16–13.74)	14.35 (12.97–16.96)	0.027	0.001	0.91	12.54 (8.45–15.53)
	CD3^+^CD16^+^CD56^+^ NK-like cells	2.14 (0.50–6.87)	4.09 (1.03–5.59)	3.34 (2.51–3.57)	0.54	0.56	0.77	2.46 (0.73–6.74)
	CD3^+^ T cells	69.70 (64.77–74.43)	71.36 (65.6–73.21)	67.57 (64.86–69.81)	0.08	0.09	0.91	69.76 (65.09–74.14)
	CD19^+^ B cells	11.73 (8.90–15.34)	11.63 (8.59–12.98)	11.21 (9.71–12.28)	0.44	0.83	0.51	11.68 (8.75–14.87)
	CD3^+^/CD4^+^ T cells	43.06 (37.02–48.81)	44.41 (41.09–48.3)	43.91 (42.42–45.60)	0.46	0.80	0.70	43.26 (37.71–48.65)
	CD3^+^/CD8^+^ T cells	26.04 (21.91–29.98)	29.22 (25.4–33.85)	34.81 (31.39–36.66)	<0.001	0.0004	0.35	26.99 (22.58–32.69)
	Ratio of CD3^+^/CD4^+^ T cells to CD3^+^/CD8^+^ T cells	1.68 (1.33–1.97)	1.565 (1.13–1.90)	1.28 (1.19–1.42)	<0.001	0.06	0.66	1.58 (1.18–1.96)
	CD4^+^PD-1^+^ T cells	29.08 (20.52–38.34)	14.14 (11.58–15.53)	5.53 (4.14–6.77)	<0.001	<0.001	<0.001	21.16 (15.47–32.76)
	CD4^+^PD-L1^+^ T cells	22.49 (14.76–27.61)	8.69 (6.20–11.30)	1.92 (1.49–2.57)	<0.001	<0.001	<0.001	19.19 (10.03–25.30)
	CD8^+^PD-1^+^ T cells	12.72 (6.76–16.76)	10.00 (7.80–10.91)	3.77 (2.38–4.63)	<0.001	<0.001	0.08	11.01 (6.90–14.62)
	CD8^+^PD-L1^+^ T cells	18.38 (15.50–19.33)	3.12 (2.57–3.66)	0.42 (0.35–0.52)	<0.001	<0.001	<0.001	15.65 (3.67–18.87)
	CD19^+^PD-1^+^ B cells	7.14 (3.97–12.1)	2.46 (1.96–4.81)	1.82 (0.80–2.44)	<0.001	0.004	<0.001	5.37 (2.53–9.63)
	CD19^+^PD-L1^+^ B cells	5.05 (3.00–11.19)	3.51 (2.62–4.18)	0.20 (0.14–0.27)	<0.001	<0.001	0.024	4.05 (2.82–8.32)

Notably, all PD-1 and PD-L1 expression parameters on T and B lymphocytes in all COVID-19 groups were clearly higher than those in controls. Greater than 10-fold differences in the expression of PD-L1 were observed between non-COVID-19 controls and ICU cases for CD4+PD-L1+ T cells (median: 1.92% [IQR: 0.98–3.91%] versus 22.49% [14.76–27.61%], respectively), CD8+PD-L1+ T cells (0.42% [0.29–0.67%] versus 18.38% [15.50–19.33%], respectively), and CD19+PD-L1+ B cells [0.20% (0.06–1.03%] versus 5.05% [3.00–11.19%], respectively). Similarly, large differences in PD-1 expression were detected between non-COVID-19 controls and ICU cases for CD4+PD-1+ T cells (median: 5.53% [IQR: 2.65–8.05] versus 29.08% [20.52–38.34%], respectively), CD8+PD-1+ T cells (3.76% [2.37–4.62%] versus 12.72% [6.76–16.76%], respectively), and CD19+PD-1+ B cells (1.82% [0.80–2.44%] versus 7.14 [3.97–12.10%], respectively). 

Furthermore, for all analysed PD-1 and PD-L1 markers, regardless of the lymphocyte subset, we observed a tendency to increase from the control group (lowest percentage) to the ICU group (highest percentage), indicating immune wasting in severely ill patients with COVID-19 (Table 2). 

We also analysed the evolution of the immunophenotype at two additional timepoints, i.e., 7 and 14 days from admission to the ICU or non-ICU unit (Table 3). Consistent PD-1 and PD-L1 expression trends were observed, with notably higher percentages of CD4+PD-1+ T cells (30.07 [21.13–36.61] versus 14.1 [11.53–15.65], respectively), CD4+PD-L1+ T cells (23.56 [14.71–29.29] versus 7.82 [5.52–10.21], respectively), CD8+PD-L1+ T cells (17.37 [12.79–18.55] versus 3.2 [2.31–3.62], respectively), and CD19+PD-1+ B cells (6.89 [4.75–12.09] versus 2.45 [1.78–4.16], respectively) among hospitalized ICU patients compared with non-ICU cases on day 7 of in-hospital treatment. Similar trends were observed on day 14 of hospitalization (26.55 [20.52–36.17] versus 14.27 [11.7–15.49]; 21.95 [15.02–28.28] versus 9.3 [6.29–10.29]; 17.69 [15.94–19.22] versus 3.31 [2.31–4.06]; 7.78 [4.31–12.46] versus 2.48 [1.56–4.95]; and 5.98 [3.06–10.08] versus 3.54 [2.42–4.56], for CD4+PD-1+ T cells, CD4+PD-L1+ T cells, CD8+PD-L1+ T cells, CD19+PD-1+ B cells, and CD19+PD-L1+ B cells, respectively; all *p* < 0.001). Receiver operating characteristic (ROC) analyses of the diagnostic accuracy of lymphocyte subsets suggest that virtually all analysed lymphocyte T and B PD-1+ and PD-L1 may be used as markers of disease severity, with variable percentages for every analysed cellular subset (Table 4). 

**Table 3 cells-11-01978-t003:** Changes in selected immunophenotype parameters during hospitalization in patients with COVID-19 on days 7 and 14 of in-hospital treatment.

Time	Parameters	Patients with COVID-19 Hospitalized in the ICU	Patients with COVID-19 Hospitalized in Non-ICU Departments	*p* Values
		Median (IQR)	Median (IQR)	
Frequencies of individual cells (%)—day 7	CD3^−^CD16^+^CD56^+^ NK cells	12.63 (7.56–16.04)	11.23 (8.22–13.74)	0.85
	CD3^+^CD16^+^CD56^+^ NK-like cells	1.66 (0.77–5.79)	2.82 (1.4–5.21)	0.52
	CD3^+^ T cells	71.73 (65.86–74.92)	69.46 (66.17–73.99)	0.66
	CD19^+^ B cells	11.22 (7.7–15.62)	9.95 (8.26–11.9)	0.54
	CD3^+^/CD4^+^ T cells	43.2 (35.95–45.69)	44.94 (40.75–48.1)	0.23
	CD3^+^/CD8^+^ T cells	26.64 (22.22–33.18)	29.52 (25.44–31.54)	0.39
	Ratio of CD3^+^/CD4^+^ T cells to CD3^+^/CD8^+^ T cells	1.62 (1.25–1.96)	1.56 (1.21–2)	0.99
	CD4^+^PD-1^+^ T cells	30.07 (21.13–36.61)	14.1 (11.53–15.65)	<0.001
	CD4^+^PD-L1^+^ T cells	23.56 (14.71–29.29)	7.82 (5.52–10.21)	<0.001
	CD8^+^PD-1^+^ T cells	11.89 (6.9–15.66)	9.97 (7.8–10.53)	0.14
	CD8^+^PD-L1^+^ T cells	17.37 (12.79–18.55)	3.2 (2.31–3.62)	<0.001
	CD19^+^PD-1^+^ B cells	6.89 (4.75–12.09)	2.45 (1.78–4.16)	<0.001
	CD19^+^PD-L1^+^ B cells	4.83 (2.29–9.96)	3.48 (2.62–4.17)	0.13
Frequencies of individual cells (%)—day 14	CD3^−^CD16^+^CD56^+^ NK cells	12.03 (7.78–15.6)	12.92 (10.5–14.61)	0.47
	CD3^+^CD16^+^CD56^+^ NK-like cells	1.8 (0.78–5.86)	1.75 (1.34–2.94)	0.95
	CD3^+^ T cells	71.88 (68.94–76.18)	71.4 (67.81–73.67)	0.20
	CD19^+^ B cells	11.43 (9.09–14.84)	10.67 (9.47–14.27)	0.85
	CD3^+^/CD4^+^ T cells	40.15 (33.08–47.31)	44.15 (42.41–47.59)	0.20
	CD3^+^/CD8^+^ T cells	27.19 (24.72–34.33)	29.06 (25.47–33.2)	0.66
	Ratio of CD3^+^/CD4^+^ T cells to CD3^+^/CD8^+^ T cells	1.43 (1.09–1.83)	1.56 (1.1–1.8)	0.77
	CD4^+^PD-1^+^ T cells	26.55 (20.52–36.17)	14.27 (11.7–15.49)	<0.001
	CD4^+^PD-L1^+^ T cells	21.95 (15.02–28.28)	9.3 (6.29–10.29)	<0.001
	CD8^+^PD-1^+^ T cells	13.57 (7.65–18.22)	9.53 (7.99–10.79)	0.08
	CD8^+^PD-L1^+^ T cells	17.69 (15.94–19.22)	3.31 (2.31–4.06)	<0.001
	CD19^+^PD-1^+^ B cells	7.78 (4.31–12.46)	2.48 (1.56–4.95)	<0.001
	CD19^+^PD-L1^+^ B cells	5.98 (3.06–10.08)	3.54 (2.42–4.56)	<0.001

**Table 4 cells-11-01978-t004:** Receiver operating characteristic (ROC) analysis of the diagnostic accuracy of lymphocyte subsets in hospitalized patients with COVID-19 (ICU versus non-ICU).

Characteristic	Prognostic Value	AUC	95% CI	Youden Index	*p* Value
Frequency of CD4^+^PD-1^+^ T cells [%] Day 0	17.52	0.99	0.96–1.0	0.91	<0.0001
Frequency of CD4^+^PD-1^+^ T cells [%] Day 7	17.43	0.995	0.98–1.0	0.94	<0.0001
Frequency of CD4^+^PD-1^+^ T cells [%] Day 14	17.35	0.97	0.92–1.0	0.88	<0.0001
Frequency of CD4^+^PD-L1^+^ T cells [%] Day 0	12.17	0.89	0.79–0.99	0.71	<0.0001
Frequency of CD4^+^PD-L1^+^ T cells [%] Day 7	11.51	0.91	0.82–0.99	0.77	<0.0001
Frequency of CD4^+^PD-L1^+^ T cells [%] Day 14	10.82	0.89	0.80–0.99	0.74	<0.0001
Frequency of CD8^+^PD-1^+^ T cells [%] Day 0	14.06	0.65	0.51–0.80	0.46	0.042
Frequency of CD8^+^PD-1^+^ T cells [%] Day 7	13.91	0.71	0.57–0.85	0.47	0.004
Frequency of CD8^+^PD-1^+^ T cells [%] Day 14	13.87	0.80	0.68–0.92	0.71	<0.0001
Frequency of CD8^+^PD-L1^+^ T cells [%] Day 0	5.76	1.00	1.0	1.0	<0.0001
Frequency of CD8^+^PD-L1^+^ T cells [%] Day 7	5.48	0.97	0.92–1.0	0.97	<0.0001
Frequency of CD8^+^PD-L1^+^ T cells [%] Day 14	6.86	0.998	0.99–1.0	0.97	<0.0001
Frequency of CD19^+^PD-1^+^ B cells [%] Day 0	5.96	0.86	0.76–0.96	0.69	<0.0001
Frequency of CD19^+^PD-1^+^ B cells [%] Day 7	6.08	0.88	0.78–0.97	0.74	<0.0001
Frequency of CD19^+^PD-1^+^ B cells [%] Day 14	5.96	0.89	0.81–0.98	0.82	<0.0001
Frequency of CD19^+^PD-L1^+^ B cells [%] Day 0	7.66	0.69	0.55–0.84	0.40	0.0068
Frequency of CD19^+^PD-L1^+^ B cells [%] Day 7	7.1	0.75	0.62–0.88	0.54	0.0002
Frequency of CD19^+^PD-L1^+^ B cells [%] Day 14	6.39	0.79	0.66–0.91	0.62	<0.0001

Importantly, all markers associated with PD-1 and PD-L1 were stable over time in the corresponding groups, with no significant variability (Figure 1).

**Figure 1 cells-11-01978-f001:**
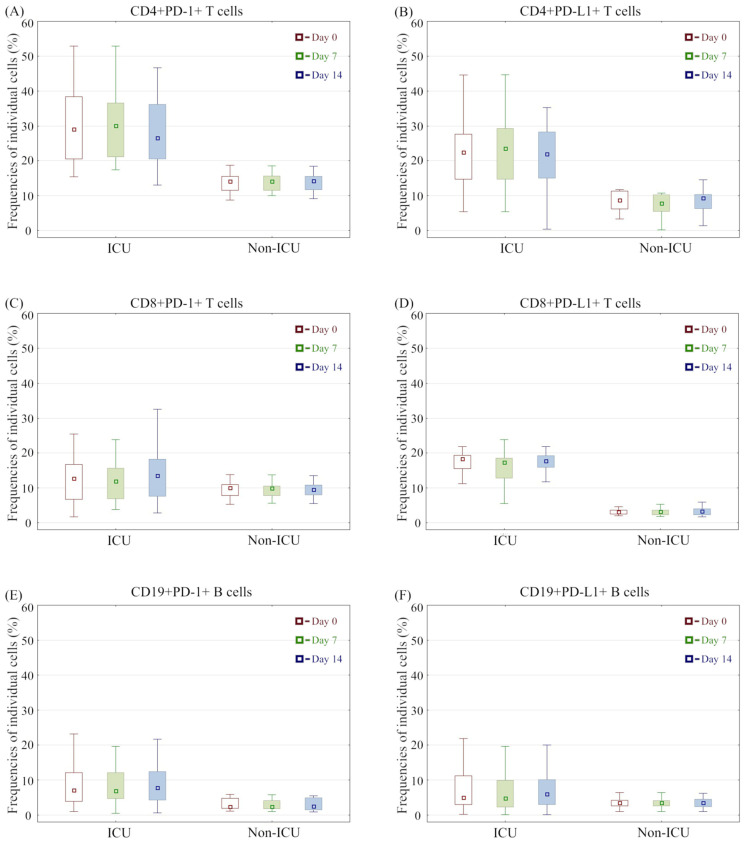
(**A**–**F**) Whisker–box plots for lymphocyte percentages over time among patients requiring mechanical ventilation (ICU group) and hospitalized patients without invasive oxygen support (non-ICU group). Medians are presented as squares inside boxes, interquartile ranges are presented as external box edges, and ranges are presented as whiskers. Percentages over time are shown for the subsequent collection days (baseline: red, day 7: green, day 14: blue).

### 3.3. Immunophenotype Differences between COVID-19 Groups and Association with Mortality

Furthermore, we divided the datasets into surviving individuals and fatal outcomes (Table 5). In these analyses, we observed increased percentages of CD4+PD-1+ T cells (27.15% [20.52%–38.47%] versus 18.16% [14.86%–29.42%], respectively) and CD4+PD-L1+ T cells (23.27% [19.18%–27.61%] versus 13.12% [8.69%–23.8%], respectively) at baseline between patients who died and survived. On day 7, there was a significant increase in CD19+PD-1+ B cells (7.32% [5.52%–12.96%] versus 4.69% [2.18%–7.03%], respectively), and on day 14, increased expression of CD4+PD-1+ T cells (25.55% [18.49%–34.38%] versus 17.65% [14.27%–27.07%], respectively), CD8+PD-L1+ T cells (17.69% [16.56%–19.85%] versus 7.26% [3.67%–17.87%], respectively), and CD19+PD-L1+ B cells (7.92% [4.05%–12.71%] versus 3.64% [2.78%–5.85%], respectively) was observed among patients with fatal outcomes compared with those in surviving patients. 

**Table 5 cells-11-01978-t005:** Comparison of selected immunophenotype parameters between surviving patients with COVID-19 and patients with fatal outcomes.

Characteristic		Patients with Fatal COVID-19	Surviving Patients with COVID-19	*p* Values
Time	Parameters	Median (Range)	Median (Range)	
Frequencies of individual cells (%)—day 0	CD3^−^CD16^+^CD56^+^ NK cells	14.97 (12.03–16.94)	10.38 (6.88–14.9)	0.035
	CD3^+^CD16^+^CD56^+^ NK-like cells	6.47 (1.03–8.41)	2.08 (0.57–5.59)	0.06
	CD3^+^ T cells	68.14 (63.76–70.58)	70.85 (65.41–74.71)	0.16
	CD19^+^ B cells	14.11 (11.73–15.34)	10.74 (7.69–13.81)	0.042
	CD3^+^/CD4^+^ T cells	41.14 (35.42–47.73)	43.8 (39.08–50.53)	0.12
	CD3^+^/CD8^+^ T cells	24.68 (20.74–29.17)	28.07 (24.54–34.42)	0.23
	Ratio of CD3^+^/CD4^+^ T cells to CD3^+^/CD8^+^ T cells	1.71 (1.43–1.96)	1.58 (1.13–1.96)	0.98
	CD4^+^PD-1^+^ T cells	27.15 (20.52–38.47)	18.16 (14.86–29.42)	0.008
	CD4^+^PD-L1^+^ T cells	23.27 (19.18–27.61)	13.12 (8.69–23.8)	0.03
	CD8^+^PD-1^+^ T cells	12.1 (6.29–14.99)	10.69 (7.8–14.25)	0.88
	CD8^+^PD-L1^+^ T cells	16.36 (14.13–19.08)	10.55 (3.22–18.81)	0.15
	CD19^+^PD-1^+^ B cells	6.65 (5.26–11.18)	4.87 (2.16–9.04)	0.07
	CD19^+^PD-L1^+^ B cells	8.03 (1.88–12.04)	4 (2.91–5.88)	0.46
Frequencies of individual cells (%)—day 7	CD3^−^CD16^+^CD56^+^ NK cells	13.11 (11.73–15.9)	10.59 (7.47–15.86)	0.07
	CD3^+^CD16^+^CD56^+^ NK-like cells	5.07 (1.08–7.16)	1.65 (0.94–5.18)	0.20
	CD3^+^ T cells	71.64 (68.18–74.55)	69.6 (65.86–74.71)	0.68
	CD19^+^ B cells	11.82 (9.66–15.99)	10.47 (7.86–12.81)	0.20
	CD3^+^/CD4^+^ T cells	44.69 (37.18–48.92)	43.11 (36.2–47.31)	0.43
	CD3^+^/CD8^+^ T cells	24.42 (20.72–28.86)	28.74 (25.31–33.04)	0.12
	Ratio of CD3^+^/CD4^+^ T cells to CD3^+^/CD8^+^ T cells	1.78 (1.45–2.15)	1.56 (1.11–1.92)	0.29
	CD4^+^PD-1^+^ T cells	25.77 (20.15–30.24)	18.83 (14.75–35.55)	0.22
	CD4^+^PD-L1^+^ T cells	20.25 (16.51–25.32)	14.47 (7.82–28.3)	0.25
	CD8^+^PD-1^+^ T cells	11.99 (6.91–15.66)	10.33 (7–13.72)	0.36
	CD8^+^PD-L1^+^ T cells	15.72 (11.86–18.37)	7.85 (3.2–17.68)	0.07
	CD19^+^PD-1^+^ B cells	7.32 (5.52–12.96)	4.69 (2.18–7.03)	0.027
	CD19^+^PD-L1^+^ B cells	8.02 (2.46–11.91)	3.78 (2.29–5.07)	0.24
Frequencies of individual cells (%)—day 14	CD3^−^CD16^+^CD56^+^ NK cells	14.89 (12.15–17.14)	10.66 (7.59–13.98)	0.003
	CD3^+^CD16^+^CD56^+^ NK-like cells	2.95 (0.78–8.19)	1.74 (0.96–4.31)	0.37
	CD3^+^ T cells	68.85 (66.88–71.8)	73.24 (70.39–75.38)	0.005
	CD19^+^ B cells	12.79 (10.56–15.8)	10.85 (9.09–14.27)	0.42
	CD3^+^/CD4^+^ T cells	39.49 (30.54–47.7)	43.73 (37.93–47.31)	0.40
	CD3^+^/CD8^+^ T cells	27.41 (23.11–34.33)	28.11 (25.38–33.2)	0.67
	Ratio of CD3^+^/CD4^+^ T cells to CD3^+^/CD8^+^ T cells	1.29 (0.97–1.9)	1.52 (1.12–1.8)	0.84
	CD4^+^PD-1^+^ T cells	25.55 (18.49–34.38)	17.65 (14.27–27.07)	0.02
	CD4^+^PD-L1^+^ T cells	21.32 (15.28–25.28)	14.52 (9.3–23.39)	0.11
	CD8^+^PD-1^+^ T cells	11.22 (6.47–15.65)	10.25 (7.76–14.72)	0.89
	CD8^+^PD-L1^+^ T cells	17.69 (16.56–19.85)	7.26 (3.67–17.87)	0.007
	CD19^+^PD-1^+^ B cells	7.76 (4.31–11.57)	4.97 (2.35–8.52)	0.06
	CD19^+^PD-L1^+^ B cells	7.92 (4.05–12.71)	3.64 (2.78–5.85)	0.039

When prognostic values calculated in ROC analysis were used as grouping factors, we found that multiple factors were also predictive of patient survival, with the strongest association observed for percentages of CD19+PD-L1+ and CD19+PD-1+ B cells on day 7 and percentages of CD4+PD-1+, CD8+PD-1+, CD8+PD-L1+ T cells, CD19+PD-1+, CD19+PD-L1+ B cells on day 14 (Table 6). 

**Table 6 cells-11-01978-t006:** Receiver operating characteristic (ROC) analysis of the diagnostic accuracy of lymphocyte subsets in the differentiation of patients with fatal COVID-19.

Characteristic	Prognostic Value	AUC	95% CI	Youden Index	*p* Value
Frequency of CD3^−^CD16^+^CD56^+^ NK cells [%] Day 0	9.49	0.71	0.53–0.88	0.51	0.022
Frequency of CD3^−^CD16^+^CD56^+^ NK cells [%] Day 14	11.98	0.84	0.70–0.97	0.64	<0.0001
Frequency of CD3^+^CD16^+^CD56^+^ NKT-like cells [%] Day 0	2.29	0.73	0.54–0.91	0.45	0.017
Frequency of CD3^+^CD16^+^CD56^+^ NKT-like cells [%] Day 7	4.51	0.70	0.51–0.89	0.43	0.035
Frequency of CD3^+^ T cells [%] Day 14	70.06	0.90	0.79–1.0	0.72	<0.0001
Frequency of CD3^+^/CD4^+^ T cells [%] Day 0	48.49	0.70	0.52–0.88	0.41	0.026
Frequency of CD4^+^PD-1^+^ T cells [%] Day 0	30.1	0.76	0.6–0.92	0.48	0.0015
Frequency of CD4^+^PD-1^+^ T cells [%] Day 14	34.3	0.82	0.67–0.96	0.65	<0.0001
Frequency of CD4^+^PD-L1^+^ T cells [%] Day 14	20.9	0.72	0.55–0.90	0.38	0.011
Frequency of CD8^+^PD-1^+^ T cells [%] Day 14	19.6	0.84	0.71–0.98	0.65	<0.0001
Frequency of CD8^+^PD-L1^+^ T cells [%] Day 14	20.1	0.91	0.79–1.0	0.83	<0.0001
Frequency of CD19^+^PD-1^+^ B cells [%] Day 7	12.96	0.87	0.74–1.0	0.74	<0.0001
Frequency of CD19^+^PD-1^+^ B cells [%] Day 14	11.1	0.89	0.78–1.0	0.67	<0.0001
Frequency of CD19^+^PD-L1^+^ B cells [%] Day 7	10.2	0.93	0.84–1.0	0.86	<0.0001
Frequency of CD19^+^PD-L1^+^ B cells [%] Day 14	15.1	0.97	0.92–1.0	0.90	<0.0001

Based on the receiver operating curves analysis, the percentage of cell subset associated with survival was assessed. The probability of survival in ICU-hospitalized patients with COVID-19 decreased significantly in groups with values > 34.3% for CD4+PD-1+ T cells, >20.9% for CD4+PD-L1+ T cells, >19.6% for CD8+PD-1+ T cells, and >20.1% CD8+PD-L1+ T cells at the 14 day time point of hospitalization (Figure 2A–D). Moreover, the frequencies of CD19+PD-1+ B cells > 13.0% and >10.2% for CD19+PD-L1+ B cells on day 7 of hospitalization in the ICU were significantly associated with fatal outcomes in patients with COVID-19.

**Figure 2 cells-11-01978-f002:**
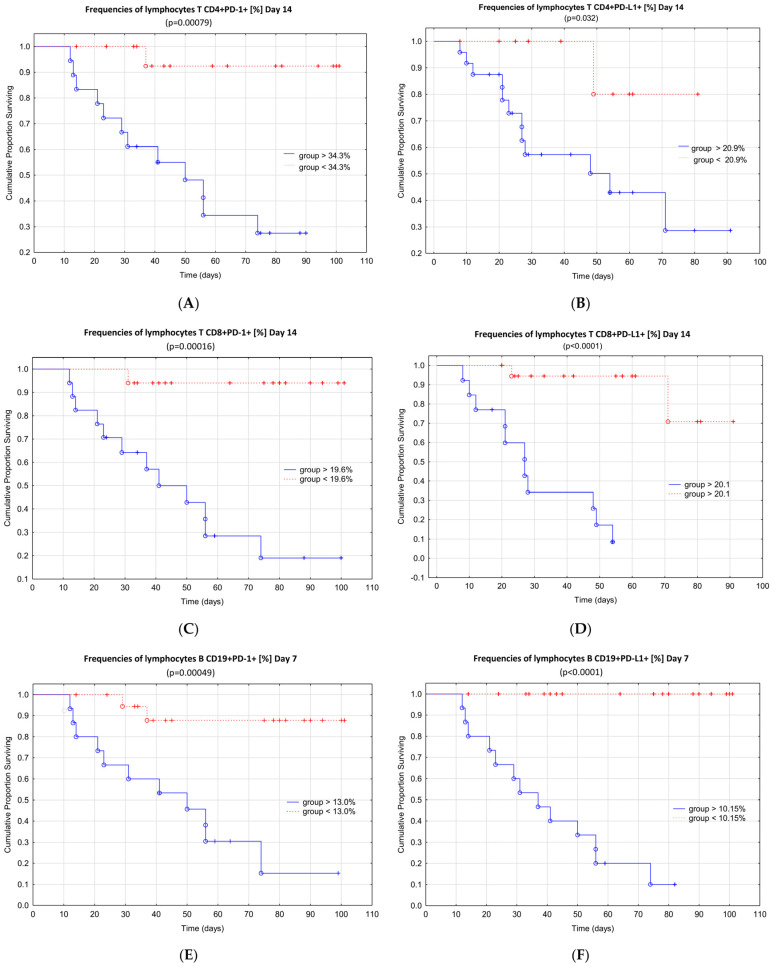
Kaplan–Meier curves illustrating the probability of survival in ICU patients with COVID-19 depending on the percentages of (**A**) CD4+PD-1+ T cells on day 14; (**B**) CD4+PD-L1+ T cells on day 14; (**C**) CD8+PD-1+ T cells on day 14; (**D**) CD8+PD-L1+ T cells on day 14; (**E**) CD19+PD-1+ B cells on day 7; and (**F**) CD19+PD-L1+ B cells on day 7. Percentage of cell subset associated with survival was selected for this analysis.

## 4. Discussion

In this study, we observed increases in immunologic parameters related to exhaustion among patients with severe COVID-19 infection, namely, PD-1 and PD-L1. The percentages of B and T cells with both these markers were notably higher among patients admitted to the ICU and in the group with fatal outcomes. Importantly, these parameters were found to be stable over time across a 14-day hospitalization period, indicating a lack of reversal of immune dysfunction among severely ill patients with SARS-CoV-2 infection. These findings suggest that exhaust-reversing agents may have therapeutic effects in patients with COVID-19-associated immune exhaustion. 

The role of the PD-1/PD-L1 axis has been studied extensively because of its potential to suppress the excessive immune response and maintain self-tolerance, thereby protecting against over-reaction of the immune system [16]. Specific monoclonal antibodies targeting the PD-1/PD-L1 signalling pathway have been used extensively in cancer immunotherapy, particularly in the treatment of metastatic melanoma, non-small cell lung cancer, head and neck squamous cell carcinoma, urothelial carcinoma, and Hodgkin lymphoma [17]. In addition, overexpression of PD-1/PD-L1 on different subsets of lymphocytes has been found to lead to the exhaustion of T cells in numerous chronic diseases, including HBV, HCV, and HIV infections, as well as a few types of latent infections, such as EBV, HSV, and VZV infections [18]. Such prolonged antigen exposure drives an “anergy” in T cells, with reduced effector function and poor proliferative capacity in exhausted populations of CD8+ T cells; the primary marker of this state is overexpression of PD-1/PD-L1 [18].

In this study, the populations of CD4+ and CD8+ T cells were notably decreased in patients with COVID-19; this effect was particularly evident in severely ill patients requiring mechanical ventilation in the ICU setting. This group was also characterised by higher expression of PD-1 receptors, which are associated with T cell exhaustion. Interestingly, increasing PD-1 expression on T cells was also observed in patients progressing from prodromal to overtly symptomatic stages, as previously described, which may be critical for the early diagnosis of patient deterioration [19]. These findings are consistent with the results from several other studies demonstrating decreases in T cell subsets, mainly CD4+ and CD8+ T cells [20,21,22,23,24]

It is worth mentioning that the COVID-19 patients included in the study were given three treatments according to the actual guidance—remdesivir (i.v.): once per day, first dose 200 mg, next 4 days 100 mg; tocilizumab (i.v.): single dose of 400/600/800 mg (depending on body weight), repeated after 8–24 h if needed; convalescent plasma (i.v.): single dose of 400 mL of AB0-compatible convalescent plasma, but these treatments did not appear to significantly impact the PD-1/PD-L1 axis.

In addition, similar to our results, all reports on the expression of PD-1 and PD-L1 in patients with COVID-19 have revealed the increased expression of checkpoint molecules on different subsets of T cells. Overexpression of PD-1 on CD4+ T cells in severe cases was detected in a study by Kuri-Cervantes et al. [25], accompanied by several changes in leukocytes, thereby confirming broad perturbations among innate and adaptive immune mechanisms. Significant reductions in T cell populations and the simultaneous overexpression of PD-1 on CD4+ and CD8+ T cells in severe cases of COVID-19 in the ICU were also confirmed by Diao et al. [19]. Moreover, similar results were observed in COVID-19 non-survivors, and increased expression of PD-1 combined with increased expression of CD38 on CD3+CD8+ T cells was shown to be a risk factor for unfavourable outcomes in patients with COVID-19 [26]. In addition, increases in soluble PD-1 in severe cases confirmed the immunophenotype trend observed in patients with COVID-19 [27]. PD-1 expression has also been shown to be increased on NK cells in critical cases [28]; however, these results have not yet been confirmed in additional studies. 

T cells, particularly CD8+ T cells, are key players in viral clearance. These cells function by secreting perforins, granzymes, and interferon (IFN)-γ to eradicate viruses from the host. Furthermore, CD4+ T cells can assist cytotoxic T cells and B cells and enhance the ability of the host to clear pathogens [29] Nevertheless, persistent and chronic stimulation by the virus leads to exhaustion, which is characterised by a loss of cytokine production and sustained expression of inhibitory receptors [29] The overexpression of PD-1 and PD-L1 in patients with COVID-19 is a strong signal of T cell exhaustion, leading to decreased T cell counts.

Based on our results, we conclude that and overexpression of PD-1/PD-L1 in CD4+ and CD8+ T cells indicated T cell functional exhaustion. However, some reports have suggested that PD-1-expressing CD8+ T cells may still be functional in patients with COVID-19 [30] based on the level of IFN-γ. Nevertheless, in our opinion, the combined immunological profile, incorporating the decrease in T cell counts together with the overexpression of PD-1/PD-L1, determine the exhaustion of T cells. This was also confirmed by Varchetta et al. [31], who characterised a wide immunological profile, including immune checkpoints such as TIM-3 and NKG2A. Exhaustion caused by increased PD-1 expression was also predicted by Zheng et al. [32] and on the basis of the T cell receptor phenotype using next-generation sequencing [33]. Overall, although decreased CD8+ T cell abundance combined with PD-1 and PD-L1 overexpression may not be sufficient to confer T cell exhaustion, CD4+ and CD8+ depletion with the overexpression of the immune checkpoint molecules may be a marker of exhaustion due to the functions of T helper cells in viral infections, e.g., sustained residual function in exhausted CD8+ T cells. Taken together, these findings suggest that without the help of functional CD4+ T cells, CD8+ T cells may be unable to react effectively. 

Study limitations include limited group sizes especially for the group of COVID-19 non-survivors; however, it is extremely difficult to collect samples for the patients with fatal outcomes, and the statistical significance for the major proportion of the analysed markers still proved strong. Furthermore, to validate immunophenotypic exhaustion parameters as markers of disease progression, larger prospective sampling from baseline with analyses of outcomes (probability of ICU admission and death) is needed. Such analysis is currently in progress. Lastly, we have selected B and T lymphocytes for the PD-1/PD-L1 marker analysis and not studied other cellular subsets such as NK cells, which is planned in the extended phase of the research. 

## 5. Conclusions

In this study, we characterised the immunological profiles of patients with different severities of COVID-19, indicating that PD-1 and PD-L1 expression are related to disease severity and mortality. Based on these results, we conclude that both PD-1 and PD-L1 analysed in various lymphocyte T and B subsets may have applications as biomarkers of COVID-19 severity in addition to its roles in cancer therapy, as previously proposed [34], and may provide a basis for the use of exhaustion reversal agents to limit the progression of the COVID-19 in the severe stages. 

## Data Availability

All date are available at Miłosz Parczewski and Paulina Niedźwiedzka-Rystwej.

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
