# Peer review of "Programmed Cell Death-1/Programmed Cell Death-1 Ligand as Prognostic Markers of Coronavirus Disease 2019 Severity"

_cells, 2022, doi:10.3390/cells11121978_

Round 1
Reviewer 1 Report
In the manuscript Niedzwiedzka-Rystwej and colleagues investigate the expression of immune cells exhaustion markers, such a PD-1 and PD-L1, in healthy donors and COVID-19 patients presenting different disease severity. The immunophenotype analyses demonstrate that modulation of PD-1 and PD-L1 expression level occur between the different sample groups and that also correlate with disease progression and mortality. Considering the pandemic situation it is of the work is of great interest however, few points should be addressed before considering it for publication:
- To better understand the results it would be better to present the immunophenotype analyses as dot plots instead of tables
- Did the authors analyse PD-1 expression on NK cells? Considering that PD-1 plays an important role also on innate immune cells it would be interesting to investigate PD-1 expression in this context
- It would be interesting to read authors’ comments on two aspects: 1) Few differences on CD8+-PD-1+ T cells expression between ICU and non ICU patients were observed compared to other cell subsets (Table 3 And Table5); however this expression seems to be correlated with patients survival (Figure 5). How authors explain these results?
Author Response
Dear Reviewer,
Thank you very much for your valuable comments in order to improve this manuscript. Here is a point-by-point response to your comments.
To better understand the results it would be better to present the immunophenotype analyses as dot plots instead of tables
RE: We decided to leave tables in the manuscript. However, according to your suggestion, we would be willing to present the immunophenotype analyses as dot plots instead of tables it in the future study. Thank you for this recommendation.
Did the authors analyse PD-1 expression on NK cells? Considering that PD-1 plays an important role also on innate immune cells it would be interesting to investigate PD-1 expression in this context
RE: We aimed to assess the significance of PD-1 and PD-L1 expression on T CD4+, T CD8+ and B cells. We did not take into account the evaluation of PD-1 positive NK cells, but thank you for the comment, we will consider this for the next study. We have added this note in the limitation of the study.
It would be interesting to read authors’ comments on two aspects: 1) Few differences on CD8+-PD-1+ T cells expression between ICU and non ICU patients were observed compared to other cell subsets (Table 3 And Table5); however this expression seems to be correlated with patients survival (Figure 5). How authors explain these results?
RE: Actually, table 3/5 and figure 5 present different analytical approaches. In the tables non-parametric statistics comparing median percentages of the analysed lymphocyte subsets are presented between the surviving and non-surviving groups, and no difference in the median percentages are observed. However, in the survival figures (figures 5) the percentage cut-off associated with survival was shown, which was based on the ROC analysis therefore these analyses differ. I have expanded the description of the KM curves below the figure 5 and also in the relevant text.
Thank you again for your kind consideration, we do hope that the manuscript can be accepted in the current form.
Kind regards,
Paulina Niedźwiedzka-Rystwej
Miłosz Parczewski
Reviewer 2 Report
The topic presented in the article submitted is interesting, well presented and well written.
In the results section the sentence in line 234 should be rephrased
Author Response
Dear Reviewer,
Thank you very much for your valuable comments in order to improve this manuscript. Here is a point-by-point response to your comments:
In the results section the sentence in line 234 should be rephrased
RE: Thank you, we have rephrased.
Thank you again for your kind consideration and good opinion on our manuscript, we do hope that the manuscript can be accepted in the current form.
Kind regards,
Paulina Niedźwiedzka-Rystwej
Miłosz Parczewski
Round 2
Reviewer 1 Report
I don't have any other comments for the authors.
This manuscript is a resubmission of an earlier submission. The following is a list of the peer review reports and author responses from that submission.
Round 1
Reviewer 1 Report
In the present manuscript, Niedźwiedzka-Rystwej et al. suggest PD1 and PD1 ligand as potential prognostic markers for COVID-19 severity. The manuscript is generally well written, and the data are interesting. However, major technical inconsistencies are present:
- How were the FACS antibodies combined to perform the FACS staining?
- Have the authors included MFO, staining and compensation controls?
- Important markers have the exact same fluorochrome. How was the analysis possible?
- The authors must include appropriate gating strategies and representative dot plots to allow a proper assessment of the results.
- Can the author indicate clones of the antibody used?
Author Response
Dear Reviewer,
Thank you for giving us the opportunity to submit a revised draft of our manuscript titled: “ Programmed cell death-1/programmed cell death-1 ligand as prognostic markers of coronavirus disease-2019 severity “ to Cells. We appreciate the time and effort that you have dedicated to providing your valuable feedback on my manuscript. We are grateful for your insightful comments on our paper. We have been able to incorporate changes to reflect most of the suggestions provided by you. We have highlighted the changes within the manuscript.
Here is a point-by-point response to your comments and concerns.
Reviewer 1
In the present manuscript, Niedźwiedzka-Rystwej et al. suggest PD1 and PD1 ligand as potential prognostic markers for COVID-19 severity. The manuscript is generally well written, and the data are interesting. However, major technical inconsistencies are present:
- How were the FACS antibodies combined to perform the FACS staining?
RE: We have expanded the methodology section and included the expemplary gating method in order to visualize the method of staining.
- Have the authors included MFO, staining and compensation controls?
RE: Compensation control has been performed according to cytometry protocol, while FMO (fluorescence minus one) was not needed due to a 2-coulur staining.
- Important markers have the exact same fluorochrome. How was the analysis possible?
RE: We have not been conducting the analysis at once, in one probe, that is how it was possible.
- The authors must include appropriate gating strategies and representative dot plots to allow a proper assessment of the results.
RE: We have introduced Figure 1 to represent the plotting strategy.
- Can the author indicate clones of the antibody used?
RE: Thank you, it is now included in the text.
Again, we would like to thank you for your time, expertise, and effort in correcting our paper, which improved due to the changes you have proposed. We hope that now it fulfills the requirements to be published in Cells.
Kind regards,
Paulina Niedźwiedzka-Rystwej
Reviewer 2 Report
The authors have presented a very important and significant topic, concerned with COVID-19 diagnostic and treatment. I would very much appreciate their effort and the depth of the study. Said that, I want to add few major and minor concerns that might potentially improve the article for the readers:
Major concerns:
- The abstract is overcrowded with numbers. No reader likes to read a paper starting in such a way. It must be rewritten in more straight-forward and lucid way.
- Introduction: What about bringing the second paragraph before the first, or some rearrangements. Just a suggestion.
- I am really curious on the effects of the three treatments given to the patients. Is there any further detailed information, the authors can provide on this? Dosing schedule of remdesivir, time of the plasma or antibody treatment. It would be an interesting observation if it is possible to categorize the data according to the treatment regimens. Just a thought. Otherwise it is not significant to include the treatments in the statistical analysis.
- Lines 183-213: It is very hard to follow the storyline due to so many numbers in the text . Rethink of better presentation and rewrite. Presenting with visuals would be great.
- Receiver operating characteristic (ROC) analysis needs to be described.
- It would be great to see the NK cells and B cells profiles plotted together with these T cells. I hope that would explore some trend about the interplay of innate and adaptive immune response on antigen presentation.
- Figure 1: Font size are small, resolution of the figure is not good enough. Rethink about the figure quality.
Minor concerns:
- Line 129: Is "ethylenediaminetetraacetic acid", the spelling correct? or is there hyphens missing? I am not sure.
- It would be easier to follow if the authors are able to add the description of the immunophenotype factors at the end of the Methods 2.4 subsection.
- Line 149: Just a simple typo: "statis-tical" should be statistical.
- I would suggest to cite the original papers of all the Methods used in the study. For example, immunophenotyping, log-rank test, Kaplan-Meyer cumulative mortality, etc.
- Table 1: Does n.s. stand for not significant? Please describe all the abbreviations used in the manuscript. Even in those n.s. cases, the authors can include the p-values and then remark them as not significant.
- Table 1: I am little confused with the total numbers of "survived" and "died". I am able to read the table correctly, total survived from non-ICU and ICU are not 35, it's 38 and total died from both cases is 14, not 17.
- Figure 2 legend is not clear. Needs more description. What is the unit of the Time axis?
Author Response
Dear Reviewer,
Thank you for giving us the opportunity to submit a revised draft of our manuscript titled: “ Programmed cell death-1/programmed cell death-1 ligand as prognostic markers of coronavirus disease-2019 severity “ to Cells. We appreciate the time and effort that you have dedicated to providing your valuable feedback on the manuscript. We are grateful for your insightful comments on our paper. We have been able to incorporate changes to reflect most of the suggestions provided by you. We have highlighted the changes within the manuscript.
Here is a point-by-point response to your comments and concerns.
The authors have presented a very important and significant topic, concerned with COVID-19 diagnostic and treatment. I would very much appreciate their effort and the depth of the study. Said that, I want to add few major and minor concerns that might potentially improve the article for the readers:
Major concerns:
1. The abstract is overcrowded with numbers. No reader likes to read a paper starting in such a way. It must be rewritten in more straight-forward and lucid way.
RE: Thank you, the abstract has been rewritten.
2. Introduction: What about bringing the second paragraph before the first, or some rearrangements. Just a suggestion.
RE: Thank you, we have used this suggestion and changed the order of the paragraphs.
3. I am really curious on the effects of the three treatments given to the patients. Is there any further detailed information, the authors can provide on this? Dosing schedule of remdesivir, time of the plasma or antibody treatment. It would be an interesting observation if it is possible to categorize the data according to the treatment regimens. Just a thought. Otherwise it is not significant to include the treatments in the statistical analysis.
RE: The treatment was performed according to actual guidance – remdesivir (i.v.): once per day, first dose 200mg, next 4 days 100mg; tocilizumab (i.v.): single dose of 400/600/800mg (depending on body weight), repeated after 8-24h if needed; convalescent plasma (i.v.): single dose of 400ml of AB0-compatible convalescent plasma. Great gratitude for your advice, we are already planning to present the impact of different treatments on PD-1/PD-L1 axis in next studies.
4. Lines 183-213: It is very hard to follow the storyline due to so many numbers in the text . Rethink of better presentation and rewrite. Presenting with visuals would be great.
RE: We have rewritten the text to make in more clear, thank you for this remark.
5. Receiver operating characteristic (ROC) analysis needs to be described.
RE: ROC has been described in the methodology section.
6. It would be great to see the NK cells and B cells profiles plotted together with these T cells. I hope that would explore some trend about the interplay of innate and adaptive immune response on antigen presentation.
RE: Unfortunately due to the technical limitations we were unable to plot those cells together, although we also think this would be interesting to see such plots. We will consider this kind of plotting in the next nearest occasion.
7. Figure 1: Font size are small, resolution of the figure is not good enough. Rethink about the figure quality.
RE: We have corrected the quality of the figure.
Minor concerns:
1. Line 129: Is "ethylenediaminetetraacetic acid", the spelling correct? or is there hyphens missing? I am not sure.
RE: WE have checked in several sources and according to them the spelling is correct. We hope this can be also verified during English editing, if the text is accepted.
2. It would be easier to follow if the authors are able to add the description of the immunophenotype factors at the end of the Methods 2.4 subsection.
RE: We have decided to expand on the section, and we added details and exemplary gating method to make the analysis easier to follow. We hope that will make the manuscript easier to follow.
3. Line 149: Just a simple typo: "statis-tical" should be statistical.
RE: Corrected, thank you.
4. I would suggest to cite the original papers of all the Methods used in the study. For example, immunophenotyping, log-rank test, Kaplan-Meyer cumulative mortality, etc.
RE: As suggested, we added the original methodology papers.
5. Table 1: Does n.s. stand for not significant? Please describe all the abbreviations used in the manuscript. Even in those n.s. cases, the authors can include the p-values and then remark them as not significant.
RE: We have added the explanation to the legend of the Table, thank you.
6. Table 1: I am little confused with the total numbers of "survived" and "died". I am able to read the table correctly, total survived from non-ICU and ICU are not 35, it's 38 and total died from both cases is 14, not 17.
RE: Thank you very much for this remark, this is obviously a mistake in the Table, which has now been corrected.
7. Figure 2 legend is not clear. Needs more description. What is the unit of the Time axis?
RE: We have expanded the description and we do hope it is more clear now which the information added.
Again, we would like to thank you for your time, expertise, and effort in correcting our paper, which improved due to the changes you have proposed. We hope that now it fulfills the requirements to be published in Cells.
Kind regards,
Paulina Niedźwiedzka-Rystwej
Round 2
Reviewer 1 Report
The research topic is interesting; however, major technical flaws are still present in the manuscript.
In FACS analyses, inappropriate gating strategy, compensation, and controls can profoundly impact the results.
The presented gating strategy is inconsistent with the material and methods and the presented data.
The authors must consider presenting a gating strategy showing a Lymphocytes'gate, the CD3 population, the CD4 and PD-1.
In addition, a gating strategy for all analysed cell types and parameters(CD8, CD19, NK...) must be included.
Representative dot plots for each measured parameter must also be included.
Author Response
Dear Reviewer,
Thank you for giving us the opportunity to submit a revised draft of our manuscript titled: “ Programmed cell death-1/programmed cell death-1 ligand as prognostic markers of coronavirus disease-2019 severity “ to Cells. We appreciate the time and effort that you and the reviewers have dedicated to providing your valuable feedback on my manuscript. We are grateful to the reviewers for their insightful comments on our paper. Here is a point-by-point response to the reviewers’ comments and concerns.
Reviewer 1
The research topic is interesting; however, major technical flaws are still present in the manuscript.
In FACS analyses, inappropriate gating strategy, compensation, and controls can profoundly impact the results.
The presented gating strategy is inconsistent with the material and methods and the presented data.
The authors must consider presenting a gating strategy showing a Lymphocytes'gate, the CD3 population, the CD4 and PD-1.
In addition, a gating strategy for all analysed cell types and parameters(CD8, CD19, NK...) must be included.
Representative dot plots for each measured parameter must also be included.
RE: Thank you for pointing out the flaws in the methodology. We have now corrected the description and added all needed Figures showing the gating strategy for all the parameters measured in the paper.
Again, we would like to thank you for your time, expertise, and effort in correcting our paper, which improved due to the changes you have proposed. We hope that now it fulfills the requirements to be published in Cells.
Kind regards,
Paulina Niedźwiedzka-Rystwej
Reviewer 2 Report
The manuscript has been improved significantly and I can appreciate the efforts made by the authors. I have few more minor comments with respect to their response to my previous concerns. With taken care of these point, the manuscript is good to be accepted.
1. Please include the future outlooks (described in the Response document) in the Discussions of the main text clearly. And also mention why you have not been able to incorporate them in the present study.
2. Another suggestion from my end is a brief of the technical workflow of the Results section before jumping into the specific study subsection 3.1 would be very helpful for the readers.
Author Response
Dear Reviewer,
Thank you for giving us the opportunity to submit a revised draft of our manuscript titled: “ Programmed cell death-1/programmed cell death-1 ligand as prognostic markers of coronavirus disease-2019 severity “ to Cells. We appreciate the time and effort that you and the reviewers have dedicated to providing your valuable feedback on my manuscript. We are grateful to you for your comments on our paper. We have highlighted the changes within the manuscript.
Here is a point-by-point response to the reviewers’ comments and concerns.
The manuscript has been improved significantly and I can appreciate the efforts made by the authors. I have few more minor comments with respect to their response to my previous concerns. With taken care of these point, the manuscript is good to be accepted.
1. Please include the future outlooks (described in the Response document) in the Discussions of the main text clearly. And also mention why you have not been able to incorporate them in the present study.
RE: We added it in the section Discussion as the Reviewer suggested.
- Another suggestion from my end is a brief of the technical workflow of the Results section before jumping into the specific study subsection 3.1 would be very helpful for the readers.
RE: Added as suggested, we do hope that we understood the intension of the Reviewer. We are ready to correct and add any amendments needed.
Again, we would like to thank you for your time, expertise, and effort in correcting our paper, which improved due to the changes you have proposed. We hope that now it fulfills the requirements to be published in Cells.
Kind regards,
Paulina Niedźwiedzka-Rystwej
Round 3
Reviewer 1 Report
The quality of the manuscript was increased. However, the description of the methodology especially the analysis of the FACS data is still not convincing and raises several concerns:
To be credible, the gating strategy must be reconsidered. For PD1 and PDL1 analyses, at least CD3, CD4 and PD1 or PDL-1 (or CD3, CD8, PD1 or PDL-1).
- According to Fig1-3, it is not clear which fluorochromes were used for the gating. Can the authors provide staining controls, including unstained and single stains? It is still unclear how proper gating can be made for all parameters using 2 colors. Even if only 2 fluorochromes (FITC and PE) were used, compensation controls are required. The FITC emission spectrum being known tooverlap the PE emission spectrum. Were the FACS data somehow compensated?
- For the SSC and FCS gating, it seems like the exact same dot plot is depicted. Have the authors used the same tube or well to set all gates?
- Have the authors used any blocking reagent during staining? Does the analysis include a doublet exclusionstrategy?
- To support their claims, the authors must include representative dot plots for each group of patients.